# The Role of Bioactive Peptides in Diabetes and Obesity

**DOI:** 10.3390/foods10092220

**Published:** 2021-09-18

**Authors:** Ramachandran Chelliah, Shuai Wei, Eric Banan-Mwine Daliri, Fazle Elahi, Su-Jung Yeon, Akanksha Tyagi, Shucheng Liu, Inamul Hasan Madar, Ghazala Sultan, Deog-Hwan Oh

**Affiliations:** 1College of Food Science and Technology, Guangdong Ocean University, Guangdong Provincial Key Laboratory of Aquatic Products Processing and Safety, Guangdong Province Engineering Laboratory for Marine Biological Products, Guangdong Provincial Engineering Technology Research Center of Marine Food, Key Laboratory of Advanced Processing of Aquatic Product of Guangdong Higher Education Institution, Zhanjiang 524088, China; ramachandran865@gmail.com (R.C.); weishuaiws@126.com (S.W.); 2Department of Food Science and Biotechnology, College of Agriculture and Life Science, Kangwon National University, Chuncheon 24341, Korea; ericdaliri@yahoo.com (E.B.-M.D.); elahidr@gmail.com (F.E.); sujung0811@gmail.com (S.-J.Y.); akanksha.tyagi001@gmail.com (A.T.); 3Southern Marine Science and Engineering Guangdong Laboratory, Zhanjiang 524088, China; 4Collaborative Innovation Center of Seafood Deep Processing, Dalian Polytechnic University, Liaoning 116034, China; 5Department of Biotechnology, School of Biotechnology and Genetic Engineering, Bharathidasan University, Tiruchirappalli 620024, India; inambioinfo@gmail.com; 6Department of Computer Science, Aligarh Muslim University, Aligarh 202002, India; ghazala.sultan2k17@gmail.com

**Keywords:** diabetes, obesity, bioactive peptides, antioxidant, anti-inflammatory, antimicrobial, antiviral

## Abstract

Bioactive peptides are present in most soy products and eggs and have essential protective functions. Infection is a core feature of innate immunity that affects blood pressure and the glucose level, and ageing can be delayed by killing senescent cells. Food also encrypts bioactive peptides and protein sequences produced through proteolysis or food processing. Unique food protein fragments can improve human health and avoid metabolic diseases, inflammation, hypertension, obesity, and diabetes mellitus. This review focuses on drug targets and fundamental mechanisms of bioactive peptides on metabolic syndromes, namely obesity and type 2 diabetes, to provide new ideas and knowledge on the ability of bioactive peptide to control metabolic syndromes.

## 1. Introduction

Bioactive peptides (BAPs) are extracted from food proteins and provide significant health effects to humans due to their immune-boosting and health-improving abilities. BAPs are proteins that exert a continuing effect beyond the established benefit of bioavailability. The ingestion of BAPs might have many positive impacts on the human body in terms of defending against pathogens [1] and modifying physiological mechanisms [2]. The sequencing of BAPs leads to their specific utilization in physiology; specifically, satiety-regulating and anti-obesity peptides serve purposes in the digestive system, the cardiac system incorporates antioxidation-facilitating, lipid-mitigating, thrombosis-inhibiting and hypertension-preventing peptides, and the immune system accepts peptides that enable immunity enhancement as well as antimicrobial and cytoplasmic regulatory peptides [3]. In vitro and in vivo studies in humans and animals have been conducted to study the health potential of food-based bioactive protein subunits with emphasis on peptides that play roles in hypertension regulation, diabetes management, cholesterol limitation, anticancer effects, and targeted microbial growth inhibition [4], as depicted in Figure 1.

BAPs, which usually contain between 3 and 20 amino acid residues, remain inactive while the sequences remain within the parent protein but become active once released via enzymatic hydrolysis by peptidases during food processing and G.I digestion. Hence, to exert a positive health effect, BAPs must cross the G.I barrier and survive enzyme degradation [5]. In recent years, numerous BAPs have been reported to be naturally present or generated from food proteins of different origins, such as milk, eggs, soybean, fish, and meat. In this sense, the bioactivity that has been most extensively studied during the last decade is antihypertensive activity, which can be assessed by measuring the angiotensin I-converting enzyme (ACE) inhibitory activity [6]. The interest in this bioactivity mainly stems from the fact that high blood pressure is a significant, independent risk factor for cardiovascular diseases and the main reason for death in developed countries. Peptides with this type of activity are generic BAPs, such as antioxidant, antimicrobial, opioid, antithrombotic, and antidiabetic peptides.

### 1.1. Sources of Bioactive Peptides

Protein-dependent peptides are predominant in animals and plants. Cattle milk and dairy foods have elevated levels of biologically active proteins, polymers, and peptides. The elevated concentrations of bioactive proteins in milk provide a good explanation for why milk is important for nutrition in babies, mostly during the golden months after childbirth [7]. Milk is ideal for the wholesome construction of nutritious peptides, and the consumption of fermented milk products that contain BAPs is advantageous, particularly in patients with hypertensive disorder [8].

Among poultry, eggs have high concentrations of BAPs that promote optimal cell health and well-being. Investigations have revealed that the antioxidant activity of egg yolk is better than that of egg white and that egg yolk exhibits more antioxidant activity than whole eggs. The boiled egg white extract would show the best bioactive function, due to the presence of strong BAPs. The main characteristic regarding meat- and fish-derived products or extracted food products is the presence of peptides that exert antioxidant, anti-inflammatory, and antiproliferative effects. Cereal grains such as barley, rice, rye, oat, millet, wheat, corn, and sorghum are a rich source of BAPs in plants. Oats and wheat contain ACE inhibitory peptides, dipeptidyl peptidase inhibitors and peptides with antithrombotic, hypotensive antioxidant, and opioid activities [9]. Rice and wheat have sequencing peptides that display anticancer activity. Among cereals, wheat and barley show the highest abundance of peptides with prospective biological activity [10]. After reviewing and selecting data from seven manuscripts, specific examples of food protein-derived BAPs and their mechanisms of action are represented in Table 1.

### 1.2. Different Functional Bioactive Peptides

#### 1.2.1. Pharmacological Properties of Bioactive Peptides

As long as the peptide retains its biochemical properties, it can be considered a bioactive drug. Peptide drug activity depends on the amino acid composition of the peptide, its N- and C-terminal amino acids, the length of the peptide chain, the charge of the amino acids, and the hydrophobic/hydrophilic nature of the amino acids. Some of the pharmacological properties of BAPs are discussed below [28].

#### 1.2.2. Antioxidant Properties of Bioactive Peptides

Oxidation is one of the major disease-causing factors in humans. Compounds that originate from dairy proteins possess anti-free-radical properties and inhibit the oxidation of fatty acids. The digestion of casein also produces phosphorylated peptides with hydrophilic and lipophilic antioxidant activity. Soy peptides offer varying degrees of hydrolysis and antioxidant activities [29].

#### 1.2.3. Antimicrobial Properties of Bioactive Peptides

Antimicrobial peptides (AMPs) inhibit microbial cell growth, such as the growth of bacteria and fungi. AMPs are polypeptides of different lengths composed of α-helical linear structures, disulphide-bridge connected cyclic structures, or open-ended cyclic structures. AMPs also have a high content of specific amino acid residues (e.g., proline-, glycine-, or histidine-rich peptides) [30].

The molecules of AMPs appear to exert cationic and hydrophobic effects by efficiently interfering with the strongly negatively charged bacterial cell wall or membrane. AMPs derived from casein exert good inhibitory effects against *Streptococcus sanguis*, *Streptococcus mutans*, *Streptococcus sobrinus*, *Porphyromonas gingivalis*, *Staphylococcus aureus*, *Salmonella typhimurium*, and *Escherichia coli* [31].

#### 1.2.4. Immunomodulatory Properties of Bioactive Peptides

Proteins and peptides from sources such as egg, milk, soy, and plant sources exhibit anti-inflammatory properties. Ovotransferrin, an egg white protein, inhibits the proliferation of mouse spleen lymphocytes. Peptides from hydrolysates of rice and soybean proteins can stimulate reactive oxygen species (ROS) and trigger nonspecific immune defence systems [32].

#### 1.2.5. Cytomodulatory Properties of Bioactive Peptides

Research studies have shown that cytoplasm-interacting compounds that target cancer cells might have anticancer effects, and the reported BAPs have been confirmed to have the potential to be effective cancer-preventive agents [33].

#### 1.2.6. Metabolic Effects of Bioactive Peptides

Metabolic changes can lead to several conditions, such as diabetes, obesity, and hypertension (characterized by elevated triglycerides, dense low-density lipoproteins, and low high-density lipoproteins). Several BAPs are involved in the functional regulation of metabolism [34]. Alpha-glucosidase and dipeptidyl peptidase IV (DPP-IV) are intimately involved in the development of type 2 diabetes (T2D). One of the peptides extracted from egg white shows antidiabetic and α-glucosidase inhibitory properties [35].

## 2. Effects of Bioactive Peptides on Human Health

Peptides are biological agents encoded inside food proteins with many beneficial and positive effects. Proteins are processed into different peptides responsible for the regulation of essential physiological natural functions, particularly the endocrinology of living organisms [36]. Moreover, these BAPs, due to their high tissue affinity, specificity, and efficiency, can interact with receptors, enzymes, and specific biomolecules in organisms and thereby induce health-promoting effects. Several studies have also revealed that these peptides exert beneficial effects in the treatment and management of chronic and several degenerative diseases, including hypertension, diabetes, obesity, and cancer [37].

### 2.1. Metabolic Effects of Bioactive Peptides

Obesity is an abnormal condition that involves excessive body fat accumulation and increases the risk of associated health problems. The worldwide prevalence of obesity has almost tripled since 1975, which is an alarming and dreadful sign of human health impairment. Obesity increases the likelihood of diseases such as type 2 diabetes, cardiovascular diseases, obstructive sleep apnoea, osteoarthritis, and depression [38]. BAPs with antioxidant, anti-inflammatory, antimicrobial, and antiviral properties have been used to treat metabolic disorders, such as type 2 diabetes and obesity. Endogenous and BAPs from various sources that contain 20 amino acid residues exhibit anti-obesity properties [39]. Analogues for novel targets such as amylin, leptin, GLP-1 MC4R, neuropeptide Y antagonists, cannabinoid type-1 receptor blockers, MetAP2 inhibitors, lipase inhibitors, and anti-obesity vaccines are currently being studied, and it is predicted that the combined use of two or more classes of drugs involving various pathways might be beneficial [40].

BAPs face several challenges affecting their prolonged use and development, and these challenges mainly include their chemical instability, hydrolysis, and aggregation due to their misfolding, short half-life, elimination, and low permeability through the cell membrane, among other characteristics. However, a peptidomimetic approach involving the editing of naturally occurring peptides is currently being used to develop promising drugs [40]. This approach includes various chemical modifications, L-to-D-form isomerization, and synthetic amino acid substitution. Researchers have aimed to target dipeptidyl peptidase-IV, STAT signalling, and protein–protein interactions for the treatment of arthritis, cardiovascular diseases, antimicrobials, immunomodulators, and peptidomimetics.

Due to the limited number of anti-obesity pharmacological drugs, chemists are now looking beyond traditional peptides and working on the development of multifunctional peptides, peptide engineering, and peptide aptamers, among others, via peptidomimetics as modern alternatives for newer design strategies [41]. These BAPs reduce the body weight of the treated animals and exert their anti-obesity activity by inhibiting the expression of the nuclear transcription factor PPARγ, as displayed in Figure 2.

### 2.2. Cholesterol-Lowering Peptides

Peptides are known to mediate the cholesterol-lowering effect of food proteins. Lupin peptides alter the transcription factors SREBP2 and HNF1α of cholesterol metabolism [42]. Phenylalanine-proline (FP) is the world’s first cholesterol-lowering dipeptide, and the cited study provided the first identification of FP as a novel cholesterol-lowering dipeptide. The active dipeptide FP can be identified by evaluations of cholesterol micellar solubility and cholesterol absorption in Caco-2 cells in vitro. However, the FP-containing fraction is not the only fraction with cholesterol-lowering properties [43].

Moreover, FP peptide might not be the only peptide in the fraction with hypocholesterolaemic properties. The oral administration of FP results in significant reductions in the total serum and non-HDL-cholesterol concentrations in hypercholesterolaemic rats fed a high-fat and high-cholesterol diet [40]. NPC1L1 is essential in the absorption of dietary cholesterol, and the mediation of NPC1L1 protein has a therapeutic function against high cholesterol [44]. Different hypocholesterolaemic BAPs and their mechanism of action are provided in Table 2. FP induces a significant decrease in NPC1L1 mRNA expression but does not significantly affect NPC1L1 expression at the protein level. Therefore, it can be speculated that the effect of FP on cholesterol absorption does not involve NPC1L1 expression [45]. ABCA1 mediates HDL cholesterol biogenesis by promoting the efflux of cholesterol and phospholipids to ApoAI22. ABCA1 is widely expressed throughout the body, as depicted in Figure 3.

### 2.3. Mechanism of Action of Anti-Diabetic Peptides against Type 2 Diabetes

T2D is a significant common human health problem that is increasing worldwide. The enzymes α-glucosidase and dipeptidyl peptidase IV (DPP-IV) play an important role in the development of T2D [57]. Hence, the reduction or inhibition of their activity might be one of the essential strategies in the management of T2D. Anti-diabetic peptides for T2D and the level of their health impacts are shown in Table 3. Studies in the field of BAPs have demonstrated that dietary proteins could be a natural source of alpha-glucosidase and DPP-IV inhibitory peptides [58,59]. Studies have shown that BAPs found in milk and other proteins play a significant role in the management of T2D through many pathways, including by decreasing appetite, regulating the plasma glucose levels, and preventing the synthesis of glucose from proteins in the body, as displayed in Figure 4 and Figure 5. Peptidyl active ingredients could be used to eliminate or decrease the incidence of T2D via the diet and supplementation.

### 2.4. Mechanism of Action of Anti-Inflammatory Peptides

Soy, which is also dense in proteins and readily available, has mainly been studied in relation to its antioxidant and anti-inflammatory activities [60]. The BAPs show anti-inflammatory activity via the NF-κB, MAPK, and/or JAK-STAT pathways by inhibiting (1) the phosphorylation of MAPKKK mediated by the interaction of the stimulus (TNFα, IL-1, or LPS) with its receptor (TNFR, IL-1R, or TLR, respectively) and (2) the downstream phosphorylation of MAPK to thus inhibit transcription factors (c-Myc, ATF-2, and c-Jun), as shown in Figure 6.

## 3. Diversity in the Production of Bioactive Peptides

Although several processes are used to generate BAPs (Figure 7), the most common method is enzymatic hydrolysis. The substrate specificity of enzymes generates peptides of different amino acid sequences. Nevertheless, peptides are complex, and a purification step is commonly employed following hydrolysis [61]. Alternatively, the synthetic production of peptides can yield specific peptides and allow the study of their physiological action.

Several research groups have used the whole hydrolysate instead of individual peptides when studying their effects [62]. In these cases, the effects cannot be attributed to a specific peptide because enzymatic hydrolysis can generate various BAPs, and thus, the observed effect could be due to a combination of numerous peptides present in the hydrolysate. Another variable of the enzymatic hydrolysis process is the processing duration, which impacts both the peptide sequences and concentration in the hydrolysate.

It is worth mentioning that the human body does not naturally produce some of the enzymes used in hydrolysate production, such as thermoase, Flavourzyme, and Neutrase, which means that the peptides produced might not replicate those generated by the natural digestion process in the human body. Some studies have used enzymes that are produced naturally in humans, such as pepsin and pancreatin. There is no guarantee that the desired peptides would be produced or stable after further GI digestion [63].

Due to the diversity of peptides obtained after enzymatic hydrolysis, the presence of multiple peptides with different mechanisms of action might influence the outcomes. The length of the peptides can affect the absorption process in the gut, as reviewed by Miner-Williams et al., and specific amino acids can significantly influence interactions with enzymes. For example, the regions of interaction between a soy peptide and the enzyme DPP-IV are correlated with the amino acids glutamine and arginine [63].

The proteolytic system of *Lactobacillus* species has the potential to produce a wide variety of peptides. *Lactobacillus* species are auxotrophic for numerous amino acids. An external source of nitrogen is needed for their growth, particularly in milk, which has low amino acid concentrations. *Lactobacillus* species have developed a proteolytic system to fulfil their nitrogen requirements, hydrolyse proteins, and supply amino acids, and this system is composed of three major components [63]. Protein hydrolysis is initiated by cell envelope proteinases (CEPs), which cleave proteins into peptides ranging from 4 to 30 amino acids. As demonstrated by gene deletion experiments, CEPs are needed for the growth of strains in milk [64]. Peptides generated by CEP cleavage are released into the extracellular media and are further internalized by different transporters, such as the oligopeptide permease (Opp) and the ion-linked transporter (DtpT) for di- and tripeptides and the ABC transporter (Dpp) for peptides containing two to nine amino acid residues. Subsequently, the internalized peptides are degraded into amino acids by the combined action of numerous internal peptidases, such as endopeptidases (PepO, PepF, PepE, and PepG), aminopeptidases (PepN, PepC, PepS, PepA, and PepL), tripeptidases (PepT), dipeptidases (PepD and PepV) and proline-specific peptidases (PepQ, PepI, PepR, PepX, and PepP) [64].

The CEPsare serine proteinases that belong to the subtilisin family. These enzymes are synthesized as preproteins of 2000 amino acids with seven functional domains (Figure 2). From the N-terminus, CEPs contain the protein prodomain (P.P.), catalytic domain (PR), insertion domain (I) (which possibly modulates the substrate specificity of the peptide), domain A (unknown function), domain B (involved in CEP stabilization), the helix domain (H) (maintains the CEPs outside the cell) and a hydrophilic domain (W; cell wall spacer or cell wall-binding domain) [65]. An anchor domain (AN) is present at the C-terminus of all CEPs with the exception of those from *L*. *Helveticus* and *L. delbrueckii* subsp. *bulgaricus* species [64].

Four different CEPs from *Lactobacillus* species have been characterized thus far, and these include PrtB from *L. delbrueckii* subsp. *bulgaricus*, PrtP from *L. casei* and *L. paracasei*, PrtR from *L. rhamnosus* and *L. plantarum* and PrtH from *L. helveticus* [65]. Although most *Lactobacillus* species possess only one CEP, the presence of four different paralogues of PrtH (called PrtH1, PrtH2, PrtH3, and PrtH4) has been observed in *L. helveticus,* and their distribution is strain-dependent [66]. The presence of several CEP paralogues, which exhibit different specificities, make *L. helveticus* the most proteolytic species among the *Lactobacillus* genus and the most efficient at generating a great diversity of “BAPs” [67]. Moreover, the proteolytic system of *L. helveticus* is the most studied due to its wide utilization for the production of dairy products, particularly cheese and fermented milk.

The proteolytic activities and protein hydrolysis patterns show marked differences among strains. These differences are likely due to multiple factors, such as differences in CEP gene expression, CEP gene mutations, and differences in optimal conditions for enzymatic activity, although the methodology used to assess the hydrolysis pattern of CEPs might also somewhat introduce variability in the reported results [68].

As expected, the differences in proteolytic activities and protein hydrolysis patterns are even more notable among different *Lactobacillus* species. The caseinolytic specificities of *L. helveticus* strains and *L. delbrueckii* subsp. *lactis* CRL581 strains are different. A similar observation was obtained from the comparison of 14 strains of *L. delbrueckii* subsp. *bulgaricus* with eight strains of *L. helveticus* [69].

These results show that *Lactobacillus* strains exhibit different hydrolytic specificities for proteins and might release very different BAPs. Some investigations are still needed to appreciate the diversity in CEP activities. A recent comparative genomic study of 213 *Lactobacillus* genomes allowed the detection of 60 different genes presenting significant homology with the currently known CEP genes (amino acid identities ranging from 100 to 20%). This high divergence level shows the potential of *Lactobacillus* species to generate a large variety of different peptides [70].

Another source of B.A.P. diversity is the type of protein used as a substrate by the bacterial strain. *Lactobacilli* can indeed hydrolyse different types of proteins, and the BAP release could be strongly dependent on the matrix. Overall, *Lactobacillus* strains are mainly used for milk fermentation, and most of the BAPs characterized thus far were isolated from milk cultures [71]. However, even milk proteins are not identical, and the same *Lactobacillus* strain can generate different peptides through the hydrolysis of caseins from cow milk, goat milk, camel milk, or mare milk. Moreover, *Lactobacillus* strains are used to hydrolyse proteins from sources other than milk, such as fish or plants, and this process can also lead to the release of specific BAPs. For example, the fermentation of pea seed by *L. plantarum* 299v leads to the release of ACE inhibitor peptides. In short, *Lactobacillus* species present great potential as producers of new BAPs due to (i) the presence of different CEPs with species- and strain-specific activities that can generate a wide variety of peptides with various molecular weights and sequences and (ii) their capacity to hydrolyse different types of proteins. Further study of the activities and specificities of CEPs will lead to a better understanding of the health potential of dietary proteins.

### 3.1. Enzymatic Hydrolysis

Enzymatic hydrolysis is a process involved in the digestion of food during which macromolecules are split from food by the enzymatic addition of water. This process has been used for sample pre-treatment in mercury speciation analysis. BAPs can be developed by the application of enzymes through hydrolysis, as presented in Table 4. The advantage of enzymolysis is that enzymes act only on specific chemical bonds. Thus, the chemical forms of mercury species are unlikely to be altered. The extract of enzymatic hydrolysis is suitable for use in reversed-phase chromatography to separate mercury species after simple filtration. Typically, the addition of small amounts of cysteine to the eluent from high-performance liquid chromatography (HPLC) can prevent the loss of mercury in the HPLC system and peak tailing and can eliminate memory effects in the inductively coupled plasma mass spectrometry (ICP-MS) system. Hydrolases have been applied to decompose the sample matrix, including protease type XIV, trypsin, lipase, and protease [72]. The significant drawbacks of enzymatic hydrolysis in sample preparation are its long sample treatment time (5–12 h) and low analyte recoveries. Ultrasonic probing can significantly accelerate the enzymatic process and shorten the extraction time to a few minutes [73].

Based on their dipeptide amino acid composition, sequences, hydrophobicity and length, peptides released from food proteins can exhibit various biological activities in addition to their nutritional properties, and these bioactivities include antihypertensive, antioxidative, antithrombotic, hypoglycaemic, hypocholesterolaemic and antibacterial activities. Protein hydrolysates are essentially produced via the enzymatic hydrolysis of whole protein sources by appropriate proteolytic enzymes under controlled conditions, and this step is followed by posthydrolysis processing for the isolation of desired and potent BAPs from a complex mixture of active and inactive peptides. Therefore, based on their human health potential and safety profiles, protein hydrolysates and biopeptides might be used as ingredients in functional foods and pharmaceuticals to improve human health and prevent diseases [84].

#### 3.1.1. In Vitro Study of Egg Hydrolysate (E.H.)/Peptides

Hen eggs and soy are two interesting sources of BAPs. Eggs are an inexpensive source of protein, are available in almost every country and are rich in nutrients, which means that they could be affordable and beneficial to a wide range of individuals around the world. Although the physiological activities of egg hydrolysate (EH) and peptides have been examined, the inhibitory, antioxidant, or anti-inflammatory effects of angiotensin-converting enzyme (ACE) have been investigated in several studies, as previously reviewed [85]. The study of eggs and soy is interesting despite their antidiabetic and anti-obesity factors because increased expression of proteins involved in the renin-angiotensin process and oxidative stress are found in diabetes and obesity, which are components of metabolic syndrome (MetS) and high blood pressure. Various in vitro studies, including assays for specific antidiabetic and anti-obesity peptides, are detailed in Appendix A. Glucose regulation involves several metabolic pathways in many organs. The inhibition of intestinal α-glucosidase, an important strategy for delaying the absorption of carbohydrates and reducing the blood glucose levels, is one approach for controlling diabetes.

Peptides obtained after egg white (EW) pepsin hydrolysis have α-glucosidase IC50 values ranging from 365 to 1694 μg/mL, whereas peptides obtained from egg yolk alcalase hydrolysis exhibit IC50 values ranging from 23 to 40 μmol/L. In addition to α-glucosidase inhibition, several activities are performed by E.W. peptides, and these include ACE inhibition with IC50 values ranging from 9 to 27 μg/mL and DPP-IV inhibition with IC50 values ranging from 223 to 1402 μg/mL. The only exception is the YIEAVNKVSPRAGQPF peptide, which exhibits no α-glucosidase or DPP-IV inhibitory effects [72]. The findings suggest that more than one physiological effect can theoretically be exerted by egg peptides. In other experiments using cell lines, several activities performed by egg white hydrolysate (EWH) have been found. Specifically, anti-inflammatory, antioxidant, hypocholesterolaemic, and DPP-IV and ACE inhibitory activities can be simultaneously exerted by EWH obtained with various enzymes. The EWH obtained via pepsin- and peptidase-mediated hydrolysis exhibits the maximum potential for MetS-related disorders such as hypertension, obesity and T2D, with IC50 values for DPP-IV and inhibition of <10 mg protein/mL and in the range of 47 to 151 μg/mL, respectively.

In a muscle cell line exposed to the EW peptides, an increase in insulin sensitivity was also observed. IRW, an egg ova transferrin peptide, strengthens angiotensin-II-induced insulin resistance in skeletal muscle cells. By normalizing the phosphorylation of the serine residue in IRS and increasing AKT phosphorylation to enhance the translocation of glucose transporter 4 (GLUT4) to the plasma membrane, the peptide reverses the impairments in insulin signalling and glucose uptake. These effects are exerted partly by minimizing angiotensin II type 1 receptor expression and reactive oxygen (ROS) generation. In contrast, the egg white-derived peptides IQW and LPK only show antioxidant activity [86].

#### 3.1.2. In Vitro Study of Egg Hydrolysate

EWH induces several biological processes in vivo in rodents. Specifically, a study found that protease-prepared EWH treatment decreases the blood glucose concentration and the insulin resistance (HOMA-IR) homeostasis model [87], does not affect the serum leptin concentrations, and reduces or does not change the plasma adiponectin levels. In addition to enlarging adipose tissue, ectopic fat accumulation can lead to insulin resistance (I.R.) and consequently T2D. An analysis of the lipid content in the liver and muscle and the total body fat percentage in rats has revealed reduced values after protease- and pepsin-prepared EWH treatment. The pepsin-prepared EWH-treated groups showed an improved steatotic state (reductions in the size and number of fat vesicles) but no histological changes in adipose tissue, as demonstrated by an adipocyte size similar to that found in the obese control group [88].

Brief descriptions of the antidiabetic and anti-obesity effects of egg-derived peptides are indicated in Appendix A. Stearoyl-CoA desaturase (SCD) is a fat synthesis enzyme responsible for converting a saturated fatty acid into its corresponding unsaturated fatty acid. The SCD index is the ratio of certain fatty acids and is related to obesity and insulin resistance. Protease-prepared EWH dietary supplementation reduces the SCD index in the serum, muscle, and liver of rodents. To elucidate the mechanisms involved in reducing fat accumulation, multiple theories have been tested; for example, SCD-1 is an enzyme important in fat synthesis, and because the EWH did not change the amount of lipogenic enzymes such as lipoprotein lipase (LPL) and fatty acid synthase (FAS), the decrease in the lipid content in non-adipose tissue has been attributed to the reduced S.C.D. index [89]. Anti-inflammatory activities, including antioxidant activities, can lead to the management of obesity and diabetes. In two in vivo trials, treatment with pepsin- and alkalase-prepared EWH reduced the plasma and kidney TNF-α level and reduced the plasma and urine malondialdehyde levels, which suggests that these peptides have antioxidant properties.

To summarize, EWH has demonstrated in vivo antidiabetic properties, as demonstrated by reductions in the accumulation of ectopic fat in the liver and muscle, improvements in insulin sensitivity and increases in the excretion of fat, which decreases the calorie intake and could lead to weight loss. EWH also protect against complications of diabetes (nephropathy), but little to no improvements in the blood glucose, adiponectin, or insulin levels, or in the inhibition of DPP-IV have been observed. The inconsistencies among the in vivo findings might be related to variations in the physiological history of the species but are more likely due to differences in the mixture of peptides present in the hydrolysates. In addition, studies have indicated that the BAPs present in the EWH are responsible for at least part of the observed effects; however, the plasma peptides were not assessed or identified, and no other particular assay was performed. As illustrated in the literature, the understanding of the absorption and action mechanism of these peptides is delayed [90].

#### 3.1.3. In Vitro Studies of Soy Hydrolysate (S.H.)/Peptides

Soybeans also contain BAPs, and Table 3 summarizes eight studies assessing the in vitro effects of SH or peptides against diabetes and obesity. Similar to EH, SH obtained via pepsin hydrolysis increases lipid accumulation, PPAR-γ expression and adiponectin expression and secretion in a dose-dependent manner during the adipocyte differentiation of 3T3-L1 preadipocytes; in addition, this SH also improves glucose uptake and GLUT4 expression, which could lead to improved insulin sensitivity. This SH is believed to stimulate preadipocyte differentiation through PPAR-γ activation, even though the SH does not exhibit PPAR-γ ligand activity [91]. HMGCoAR, 3-hydroxy-3-methylglutaryl CoA reductase; LDL, low-density lipoprotein; LDLR, LDL receptor; PCSK9, proprotein convertase subtilisin/kexin type 9; SREBP2, sterol regulatory element-binding protein 2; HNF1a, hepatocyte nuclear factor 1a.

Higher lipolysis in mature 3T3-L1 cells has been observed after treatment with Flavourzyme-prepared S.H. even after GI -simulated digestion. Along similar lines, SH prepared with alcalase lowers lipid accumulation and down-regulates LPL and FAS gene expression (enzymes involved in lipid uptake and de novo fatty acid synthesis) in the absence of or following GI-simulated digestion. A hydrolysate obtained only with naturally occurring enzymes (pepsin + pancreatin) exerts similar effects, although to a lesser extent. The findings suggest that GI digestion in vivo might not markedly affect the bioavailability of S.H., although whether this is true for all hydrolysates remains to be determined. β-Conglycin is a storage protein naturally found in soybean. Interestingly, a higher β-conglycin concentration in the hydrolysate is related to higher LPL and FAS inhibition [92].

In addition, the peptides increased glucose uptake and strengthened the liver cell expression of GLUT4 and GLUT1. DPP-IV inhibitory activity with an IC50 value of 106 μM was also observed with one of these peptides (IAVPTGVA), and the regions of interaction between IAVPTGVA and DPP-IV were identified as the amino termini of residues Glu205 and Glu206 and the carboxyl terminus of residue Arg358. Inflammation was not the subject of this study but is associated with diabetes and obesity. In co-cultured adipocytes and macrophages, which exhibit reductions in COX-2 and inducible nitric oxide synthase protein, two studies have documented changes in inflammatory markers after treatment with S.H. or soy peptide. These treatments decrease the output of nitric oxide and prostaglandin E2. Synthetized soy peptide FLV treatment decreases the production and effect of inflammatory molecules and increases insulin sensitivity in adipocytes (higher phosphorylation of IRS-1 and AKT) [30]. The authors showed evidence demonstrating that the transport of peptides into 3T3-L1 cells mainly occurs through the peptide transporter PepT2 [93]. Certain diabetes- and obesity-related soy-derived peptides and their outcomes and particularly their enzymatic activity are detailed in Appendix A.

#### 3.1.4. In Vivo Studies of Soy Hydrolysate (S.H.)/Peptides

In vivo studies have demonstrated that S.H. modulates glucose metabolism and reduces body weight. The 37-amino-acid soy peptide aglycin improves muscle glucose uptake by increasing the phosphorylation of the insulin receptors IRS-1 and AKT and enhancing the membrane GLUT4 levels, which contributes to improved insulin sensitivity in mice with T2D. Treatment with aglycin yielded results similar to those obtained with metformin in oral glucose tolerance (OGTT) and insulin tolerance tests; furthermore, the release of insulin during OGTT was normal in the treated animals and, as expected, abnormal in the T2D mouse controls, which suggests that the effect on glucose tolerance was primarily due to enhancements in glucose uptake and insulin sensitivity. Notably, intact aglycin (37 amino acids) was found in blood samples from mice, which indicated that this peptide is stable after GI digestion and is likely absorbed intact [94].

Protease-prepared S.H. decreases fat accumulation in genetically obese mice and increases lipid excretion and enhances the plasma CHOL levels in rats with diet-induced obesity with respect to the serum lipid profile and lipid excretion. The decrease in fat accumulation might be attributed to the greater postprandial energy expenditure observed after protease-prepared SH intake relative to that observed with casein; in addition, an increase in exogenous carbohydrate oxidation was the major contributor to the increased postprandial energy expenditure. Although the impact on energy expenditure was not maintained after 24 h, total carbohydrate oxidation continued to be higher in the SH-treated group. This finding might be due to higher levels of plasma insulin, lower glucose levels, lower lipid absorption, and increased absorption of carbohydrates during the postprandial era, but no studies have been performed to substantiate these conclusions [29].

### 3.2. Gastrointestinal Digestion

Essentially, the GI tract is a tube that stretches from the mouth to the anus and typically has the same composition throughout. The tube has a hollow part known as the lumen, which consists of a layer of epithelial cells and a muscle layer in the centre. The preservation of the mucosal integrity of the tract is the responsibility of these layers. Many hormones and digestive enzymes are secreted into the organs of the GI system. This section explores the origin and function of major GI hormones such as gastrin, ghrelin, cholecystokinin (CCK), and secretin. Eating allows the secretion of GI hormones, and some of these hormones, which are known as incretins, facilitate plasma glucose-lowering effects through the secretion of insulin from pancreatic β-cells. Incretin mimetics are currently used for the treatment of T2D. GI hormones regulate satiety and appetite at the CNS level in addition to their digestive function and play an essential role in energy homeostasis [95].

### 3.3. Fermentation

Over the past hundred years, the word “fermentation” has experienced several shifts in context. According to its derivation, “fermentation” merely means a gentle bubbling or boiling state. When the only known reaction of this type was wine production, bubbling triggered by carbon dioxide production was the first application. It was not until the chemical aspects of the process were analysed by Gay-Lussac that the definition was altered to indicate sugar breakdown into ethanol and carbon dioxide. However, Pasteur’s identification of microbes involved in fermentation in 1857 marked the beginning of chemical microbiology. In referring to the microbes, Pasteur interchangeably used the words “cell” and “ferment”. Therefore, the term “fermentation” has been associated with the definition of cells, the production of gas, and the production of organic byproducts [96]. Fermented foods can be defined as products with physical, chemical, and biological characteristics that have been changed by the action of microorganisms. These products contain various microbial metabolites, such as alcohol, lactic acid, propionic acid, acetic acid, exopolysaccharides, and carbon dioxide, as well as refined molecules derived from the original foodstuff. These derived products, which could be called “tertiary metabolites,” might play an important role in the biological activities of the fermented product. A successful demonstration of this process is the fermentation of milk by lactic acid bacteria (LAB) [45].

Fermentation is the anaerobic decomposition of organic compounds into organic products that the cells’ enzyme systems cannot metabolize without oxygen intervention. The fermentation results vary among various microorganisms and are primarily regulated by the cells’ dynamic enzymes and the environmental conditions. The economic value of these byproducts has led to the development of industrial microbiology. Indeed, dietary proteins are among the most potent food sources of physiologically active molecules. Their degradation by proteolytic enzymes during the digestion process releases small- or medium-sized peptides whose structures can be similar or very close to those of human or animal hormones. As some of these peptides might pass through the intestinal mucosa at this stage [97] or interact with biological receptors, it is clear that some biological responses might be triggered or intensified. Several bioactive peptide fractions extracted through enzymatic proteolysis from milk proteins have also been isolated and identified. These physiologically active peptides include opiates, angiotensin-converting enzyme inhibitors, platelet aggregation inhibitors, antibacterials, digestive system regulators, and immune modulators [98].

### 3.4. Genetic Engineering

Genetic engineering involves the use of different techniques to intentionally modify genetic material (primarily deoxyribonucleic acid or DNA) to alter, restore, or boost the shape or function. Recombinant DNA technologies developed in the latter half of the twentieth century included the application of either bacteria (such as Escherichia coli) or bacteriophages (viruses that infect bacteria, such as λ phage) or direct microinjection via the chemical splicing (recombination) of various DNA strands. In recent years, modern techniques to develop and engineer new life forms, which are commonly referred to as synthetic biology, have replaced these conventional methods. Genetic engineering raises several significant ethical issues. For instance, in agriculture, ethicists have highlighted potential human health hazards associated with genetically modified crops and livestock as well as normative concerns regarding the treatment of animals and the ecological consequences of genetic engineering. There is also substantial ethical controversy regarding the putative distinction between protocols meant to restore function and those meant to enhance function beyond species-typical norms in medicine. Additionally, ethicists have addressed the potential human health risks associated with germline genetic engineering, which is distinct from somatic genetic engineering. Finally, in the context of reproduction, ethicists have argued that genetic engineering raises ethical issues involving the screening and manipulation of embryos to eliminate or introduce various medical and cosmetic characteristics.

Genetic engineering poses additional ethical concerns regarding its potential social ramifications and the wisdom of plant, animal, and human genetic manipulation. Public health programmes have historically aimed to improve sanitation, confirm the availability of clean water, recognize the source of infectious disease vaccines and develop them in pursuit of objectives related to health promotion and disease prevention. However, the scope of potential public health strategies has significantly expanded with the advancement of genetic modification technologies and the sequencing of plants and animals (including humans), but the risks to public health have also increased [99].

## 4. Purification and Characterization of Bioactive Peptides

Peptides are molecules of essential value, and many elicit cellular responses. Different purification methods for BAPs are presented in Figure 8 based on their source. Many new genes have been discovered through the completion of several genome sequencing projects. However, because functional peptides also contain posttranslational modifications occurring at different lengths, it is of great importance to detect, purify, and classify novel BAPs, as displayed in Figure 8.

New methods need to be developed for the identification, isolation, and functional characterization of peptides [100]. Peptides typically have 3–20 amino acid residues, and their bioactivity is dependent on their compositions and sequences of amino acids. Recent studies have shown that following their release via enzymatic hydrolysis, most peptide sequences encrypted in food proteins confer bioactive properties. The peptide structure is essential for recognition; thus, many researchers have investigated the purification of peptides. Peptides were first collected from marine organisms using a standard technique for detecting marine BAPs. The extract was screened for special bioactivity, fractionated using fractionation technology driven by a bioassay, and finally purified to generate a single bioactive peptide. Study techniques such as membrane filtration systems, gel or size-exclusion chromatography, and capillary electrophoresis (C.E.), as shown in Figure 9, ion-exchange column chromatography as depicted and reverse-phase high-performance liquid chromatography (RP-HPLC) as displayed in Figure 10, are important for establishing an effective purification method. Each purification technique has advantages and disadvantages that researchers should carefully consider before purifying peptides [101].

## 5. Conclusions

This narrative review provides an overview of agro- and farm-based products that can be a rich source of BAPs, their potential physiological activities, such as anti-obesity and antidiabetic effects, and their contribution to the management of other metabolic disorders. The production of BAPs can be achieved enzymatically, chemically, or through a process based on fermentation, which leads to the hydrolysis process and the development of a novel combination of peptides.

In addition, the period of treatment varies depending on the different physiological and genetic backgrounds, such as obesity, diabetes, and normal physiology, and thus, a standardized physiological model is needed to evaluate the activity of the peptides. Despite these limitation, optimization of the dosage using a various clinical trial of soy and egg-based bioactive peptics could overcome the hurdles and yield an efficient treatment for obesity and diabetes.

Despite all clinical trials performed, the mechanisms of absorption and action of the peptides in cell-level metabolism, such as the molecular target insulin and glucose metabolism, the different soy varieties show varied efficacies. Thus, more studies are needed to elucidate the effect of soy and egg BAPs.

## Figures and Tables

**Figure 1 foods-10-02220-f001:**
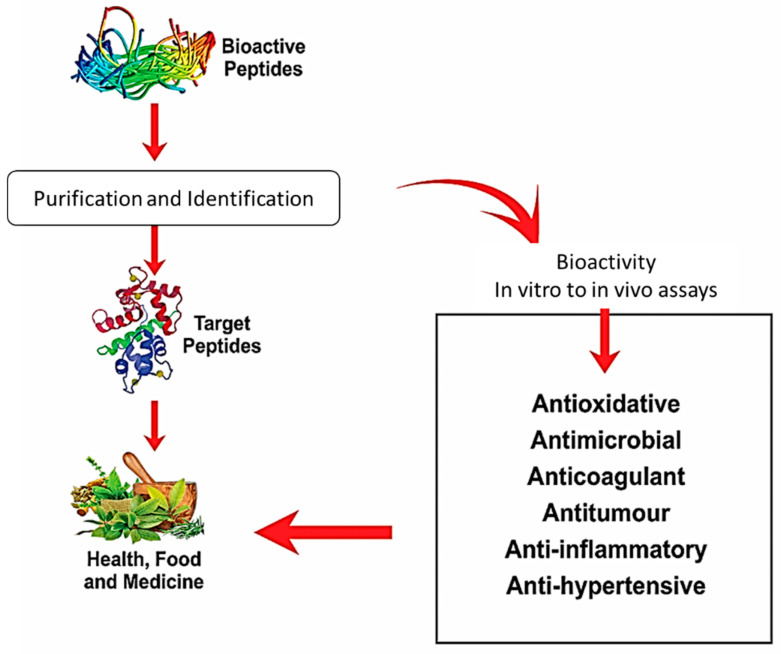
Bioactive peptides and associated health benefits, where attributed to their physiological activities exerted in vitro and in vivo, these include lowering blood pressure by inhibiting the angiotensin-converting enzyme (ACE); oxidative stress reduction by neutralizing or scavenging free radicals; antimicrobial (targeting binding efficacy towards the pathogen cell wall), anticoagulant, antitumor (targeting cancer cells), and anti-inflammatory (triggering the anti-inflammatory cytokines).

**Figure 2 foods-10-02220-f002:**
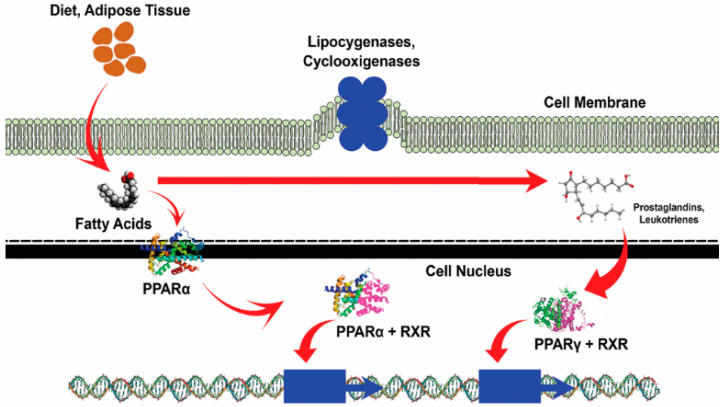
Proposed model of the cross-talks between PPARs and bioactive peptide in obesity. The anti-obesity activity of bioactive peptide by downregulating the expression of the nuclear transcription factor PPARγ.

**Figure 3 foods-10-02220-f003:**
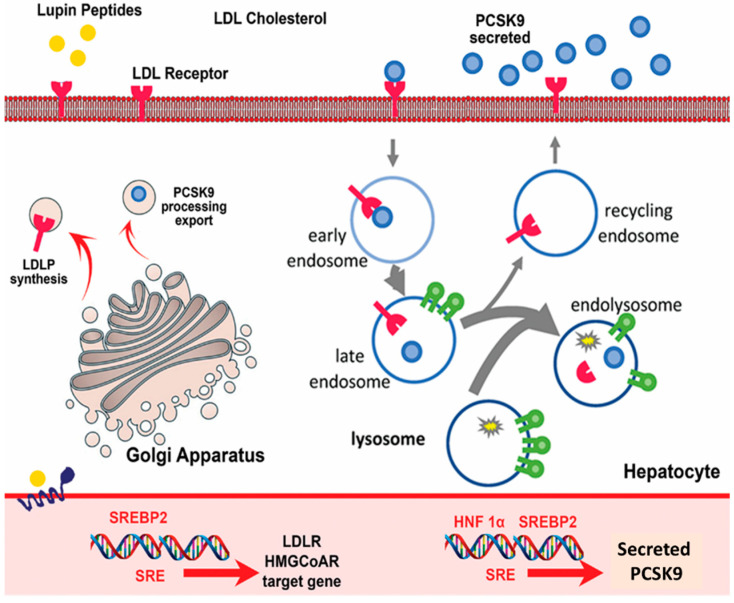
Mechanism of action of cholesterol-lowering peptides indicates the summary of the hypocholesteromic mechanisms of lupin protein-derived peptides in hepatocytes. Lupin protein represent the cholesterol-lowering effects targeting PCSK9: from clinical evidence to elucidation of the in vitro molecular mechanism using HepG2 cells.

**Figure 4 foods-10-02220-f004:**
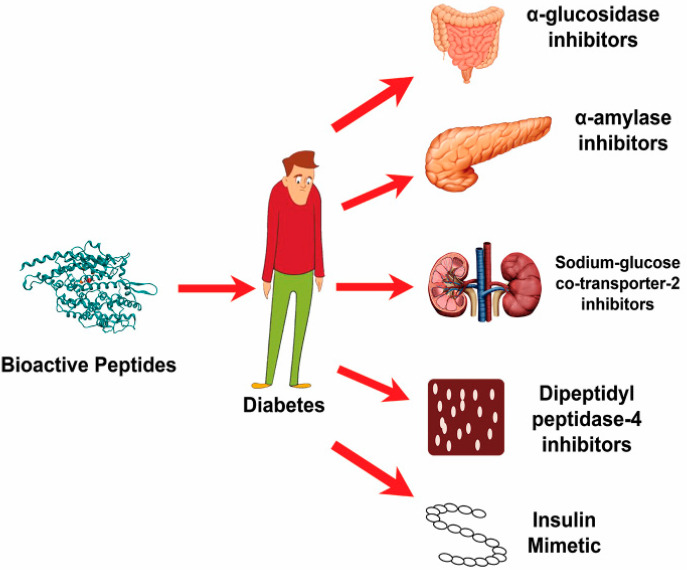
Bioactive peptides exhibit antidiabetic effects for type 2 diabetes mellitus based on inhibition against α- amylase, α-glucosidase, sodium glucose co-transporter-2 inhibitors, plasma-based dipeptidyl peptidase-4 (DPP4) inhibitors (an obesity-independent parameter for glycaemic deregulation in type 2 diabetes patients), and insulin mimetic (which promote the glucose entry into the tissues, whereas the glucose either be converted into energy or stored for later use), respectively.

**Figure 5 foods-10-02220-f005:**
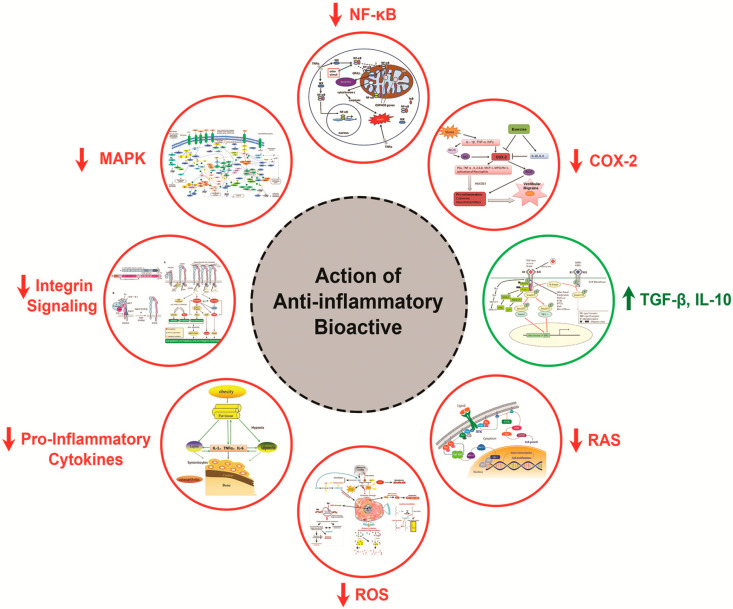
The schematic diagram indicates the anti-inflammatory activity of bioactive peptides derived from food protein occurs via inhibition of the NF-KB, MAPK, and JAK-STAT pathways. MAPK: mitogen-activated protein kinase; MAP3K: MAPK kinase; NF-κB: nuclear factor-kappa B; TGF-β: transforming growth factor β; TNF-α: tumor necrosis factor α; JAK-STAT: Janus kinase-signal transducer and activator of transcription.

**Figure 6 foods-10-02220-f006:**
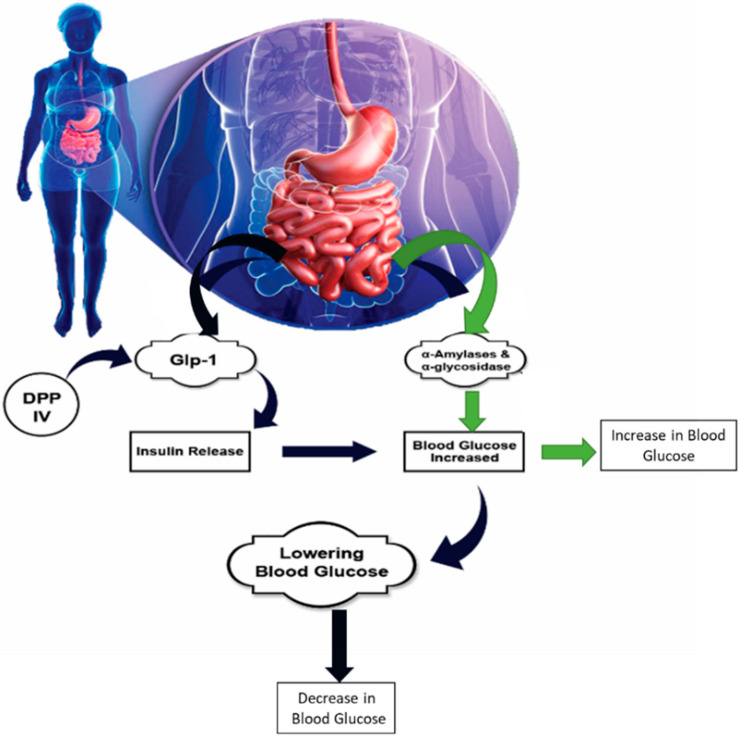
Bioactive peptides, inhibit key enzymes involved in diabetes—DPP IV, α-amylase, and α-glucosidase, which results in the antidiabetic activity mainly by promoting insulin signaling and the AMPK signaling pathway.

**Figure 7 foods-10-02220-f007:**
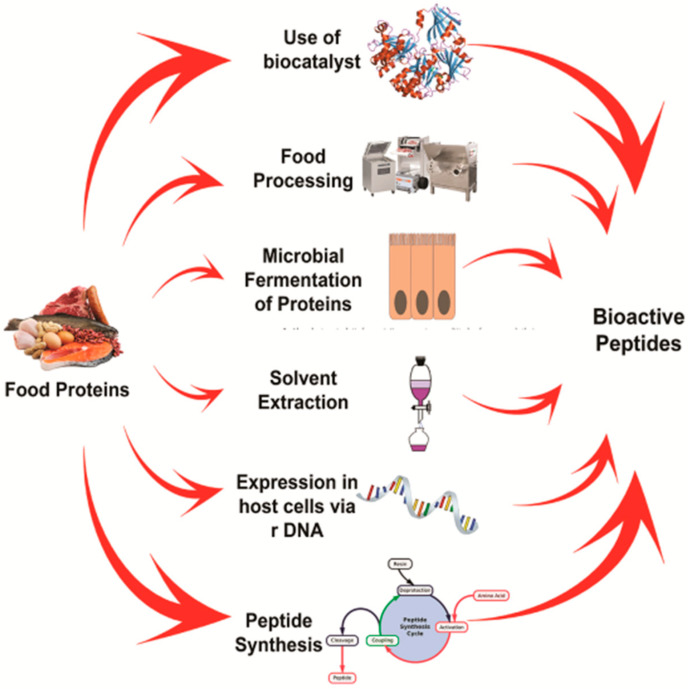
Production of bioactive peptides based on the enzymatic hydrolysis (using proteolytic enzymes from either plants or microbes), hydrolysis with digestive enzymes (simulated gastrointestinal digestion), by fermentation using starter cultures, solvent extraction based on the precipitation of the proteins, triggering the overexpression of the peptide based on stress mechanism.

**Figure 8 foods-10-02220-f008:**
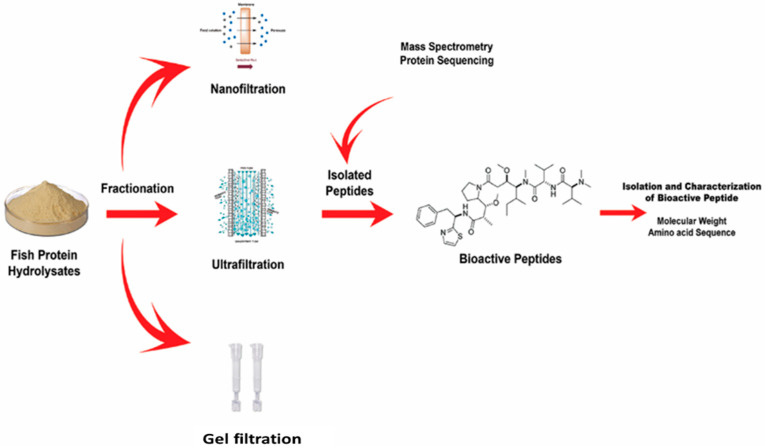
Structural characterization of fish-based bioactive peptides was determined initially by purification through nano-filtration, ultra-filtration, and gel-filtration; the purified peptides were further characterized based on the molecular weight using high-performance liquid chromatography mass spectrometry and the protein sequence was determined using liquid chromatography mass spectrometry.

**Figure 9 foods-10-02220-f009:**
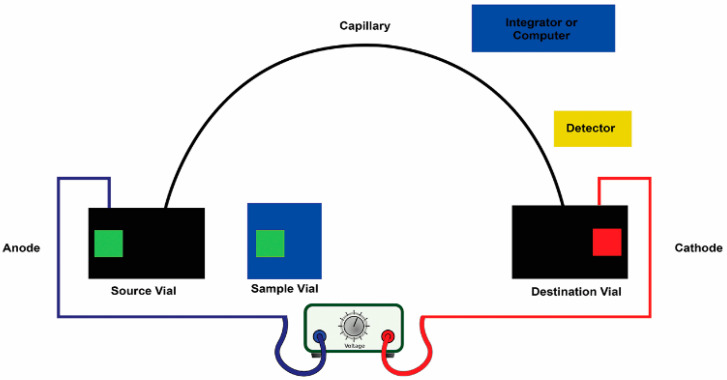
Protein mixtures can be characterized in terms of their separations by capillary electrophoresis (CE). The detection of peptide in CE is usually based on the ultraviolet (UV) absorbance of the peptide bond at or near 200 nm.

**Figure 10 foods-10-02220-f010:**
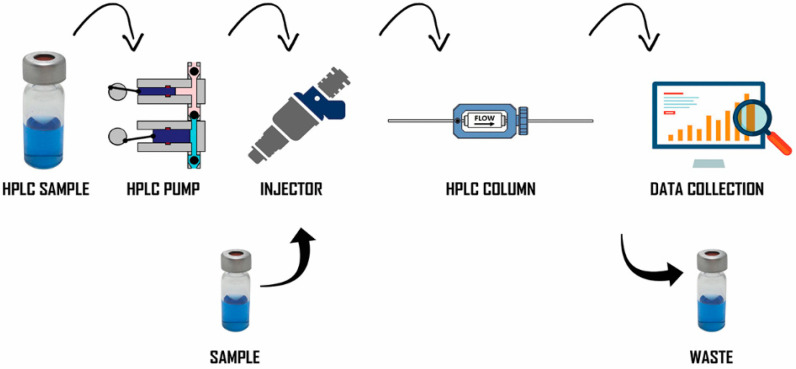
Reversed-phase high-performance liquid chromatography (RP-HPLC) involves the separation of molecules on the basis of hydrophobicity. The separation depends on the hydrophobic binding of the solute molecule from the mobile phase to the immobilized hydrophobic ligands attached to the stationary phase. (1) The resolution achieved under a wide range of chromatographic conditions, even a closely related peptides and structurally quite distinct peptides; (2) selective chromatographic was obtained based on altering the mobile phase characteristics; (3) leads to higher level of peptide recoveries leads to increased productivity; and (4) the reproducibility on the separations were stabile under a wide range of mobile phase conditions.

**Table 1 foods-10-02220-t001:** Examples of food protein-derived bioactive peptides and their mechanisms of action. ABCB-1: ATP-binding cassette subfamily B member 1; ACE: angiotensin I-converting enzyme; Ang II: angiotensin II; AT1R: angiotensin II type-1 receptor; CDH-1: cadherin 1; eNOS: endothelial nitric oxide synthase; CCK1: cholecystokinin-1; COX: cyclooxygenase; IRF-8: interferon regulatory factor 8; PGI2–IP.: prostacyclin receptor; ICAM-1: intercellular adhesion molecule-1; SOD: superoxide dismutase; SHR: spontaneously hypertensive rat.

Peptide	Source	Animal	Activity	Reference
Peptides with antihypertensive effects: RRWQWR, IJWKL and RPYL	Milk, Derived from lactoferrin	SHRs	Reduces Ang II-induced vasoconstriction in isolated rabbit carotid arterial segments.	[11]
IRW	Derived from egg ovotransferrin	SHRs	Dose-dependently attenuates BP by ~10 mmHg and ~40 mmHg in the low- and high-dose groups, respectively, compared with the results obtained with untreated SHRs. Increases the expression of ACE-2, ATP-binding cassette subfamily B member 1, interferon-regulatory factor and cadherin 1 while significantly decreasing the expression of ICAM-1 and VCAM-1 in mesenteric arteries.	[12,13]
EWH (egg white protein hydrolysate)	Egg white protein	SHRs	Improves vascular relaxation and modifies aortic Ang II receptor expression.	[14]
YRGGLEPINF ESIINF	Egg white protein	SHRs	Its vascular-relaxing mechanism is independent of ACE inhibition.	[15]
TRB (Thermolysin- digested rice bran) LRA, YY	Thermolysin-digested rice bran	SHRs	The long-term administration of TRB (50 mg kg^−1^ d^−1^) lowers systolic blood pressure compared with that of the control group. TRB reduces ACE activity in the lung in a dose-dependent manner but does not affect ACE activities in the aorta, kidney, and heart tissues. LRA (0.25 mg kg − 1) and YY (0.5 mg kg − 1) lowers blood pressure 4 h after oral administration.	[16]
IQP, VEP	*Spirulina platensis* hydrolysates	SHRs	Inhibits ACE, Ang II and AT1R. Upregulates ACE2, Ang (1–7), Mas and AT2.	[17]
Salmon gelatine hydrolysate	Salmon	SHRs	ACE and DPP-IV inhibitory activities in vitro.	[18]
SLR, YY, ER, and FR	Papain-digested bovine serum albumin	SHRs	ACE inhibitory activity in vitro and in vivo.	[19]
Hydrolysate fraction	Palm kernel	SHRs	Antihypertensive effects.	[20]
Milk peptides	Milk protein hydrolysate	SHRs	Attenuate the development of hypertension: systolic blood pressure is increased 33 ± 3 mmHg in the control group compared to 18 ± 5 mmHg in the treated group. Improve aorta and mesenteric acetylcholine relaxation. Increase eNOS expression in the aorta. Decrease left ventricular hypertrophy and interstitial fibrosis.	[21]
RVPSL	Egg	SHRs	Dose-dependently decreases systolic blood pressure starting one week after the administration of a maximum dose of 50 mg/kg. Increases the mRNA expression of renin, ACE, and AT1 receptor in the kidney. Decreases the serum Ang II, renin, and aldosterone levels.	[22]
Rapakinin (RIY)	Rapeseed	SHRs	Induces dilatation of mesenteric artery mediated mainly by the PGI2–IP receptor and then CCK–CCK1 receptor-dependent vasorelaxation.	[23]
Other effects VPP	Milk, casein-derived	CSJEL/61 Mice	Attenuates high-fat diet-induced adipose tissue inflammation.	[24]
Hemp seed Protein hydrolysate	Hemp seed	SHRs	Decreases SOD and catalase expression and the total peroxides levels.	[25]
Milk peptides	Fermented milk with *Lactococcus lactis* NRRL B-50571	SHRs	Enhance nitric oxide production and antioxidant activity.	[26]
Lunasin	Soy	Apolipoprotein E–deficient (ApoE-/-) mice	Decreases plaque formation in an experimental ApoE2/2 atherosclerotic model.	[27]

**Table 2 foods-10-02220-t002:** Cholesterol-lowering peptides. Mechanisms underlying the hypocholesterolaemic activity of peptides.

Peptide Source	Peptide Sequence	Hypocholesterolaemic Mechanism	Reference
Cowpea	Peptide mixtures	Binding to bile acids/salts or lipids and inhibition of micellar cholesterol solubility	[42,43,44,45,46]
Sericin	Peptide mixtures		
Royal jelly	Peptide mixtures		
Chemical synthesis	KRES		
Rice bran	Not applicable		
Cowpea	Peptide mixtures		
Lupin	Peptide mixtures		
Hempseed	Peptide mixtures		
Amaranth	GGV, IVG, VGVL		
Soy b-conglycinin	YVVNPDNDEN	Inhibition of HMGCoAR activity and inhibition of the mevalonate pathway and cholesterol biosynthesis	[44,47,48,49,50,51]
Soy b-conglycinin	YVVNPDNDEN		
Soy glycinin	IAVPGEVA		
Soy glycinin	IAVPTGVA		
Soy glycinin	LPYP		
Lupin b-conglutin	LILPKHSDAD		
Lupin b-conglutin	LTFPGSAED		
Chemical synthesis	P.M.A.S.		
Lupin	Peptide mixtures		
Hempseed	Peptide mixtures		
Soy glycinin	IAVPGEVA	Increases in the SREBP2 and L.D.L.R. protein levels and increases in LDL uptake and cholesterol degradation	[47,48,52]
Soy glycinin	IAVPTGVA		
Soy glycinin	LPY.P.		
Soy b-conglycinin	YVVNPDNDEN		
Soy b-conglycinin	YVVNPDNDEN		
Lupin proteins	Peptide mixtures	Decreases in PCSK9 production (via an effect on HNF1a protein) and secretion and increases in the LDLR level and the uptake of LDL by hepatocytes	[53,54]
Lupin b-conglutin	LILPKHSDAD	Inhibition of PCSK9–L.D.L.R. interaction and increase in LDL uptake	[55,56]

**Table 3 foods-10-02220-t003:** Antidiabetic peptides for type 2 diabetes mellitus (T2DM).

Antidiabetic Peptides	Hypoglycaemia	Weight Gain	Oedema	GI Effects	Lactic Acidosis	Liver Toxicity
Glipizide XL	1+	1+	0	±	0	±
Glyburide	4+	2+	0	±	0	±
Glimepiride	2+	1+	0	±	0	±
Repaglinide	1+	1+	0	0	0	0
Nateglinide	1+	↓	0	0	0	0
Metformin	0	?	0	2+	1+	0
Acarbose	0	0	0	3+	0	±
Miglitol	0	0	0	3+	0	0
Rosiglitazone	0	3+	2+	0	0	0 *
Pioglitazone	0	3+	2+	0	0	0 *

0 = none, ± = very infrequent, 1+ = infrequent, 2+ = occasional, 3+ = moderate, 4+ = significant, ↓ = decrease, ? = unknown. * The liver function monitoring recommendation was adapted from [4] articles published under an open access Creative Common CC BY license, MDPI, 2018.

**Table 4 foods-10-02220-t004:** Enzymes applied in the production of bioactive peptides by enzyme hydrolysis.

Microorganisms Used	Precursor Protein ^a^	Peptide Sequence	Bioactivity	References
*Lactobacillus helveticus*, Sacch *aromyces cerevisiae*	β-cn, k-cn	Val-Pro-Pro, Ile-Pro-Pro	ACE inhibitory and antihypertension activities	[74,75]
*Lactobacillus* G.G. enzymes + pepsin and trypsin	β-cn, αs1-cn	Tyr-Pro-Phe-Pro, Ala-Val-Pro-Tyr-Pro-Gln-Arg, Thr-Thr-Met-Pro-Leu-Trp	Opioid and ACE inhibitory activity immunostimulatory activity	[76]
*L. helveticus* CP90 proteinase	β-cn	Tyr-Pro-Phe-Pro, Ala-Val-Pro-Tyr-Pro-Gln-Arg, Thr-Thr-Met-Pro-Leu-Trp	ACE inhibitory activity	[77]
*L. helveticus* CPN 4	Whey proteins	Tyr-Pro	ACE inhibitory activity	[78]
*L. delbrueckii* subsp. *bulgaricus* SS1, *Lactococcus lactis* subsp. *cremoris* FT4	β-cn, k-cn	Many fragments	ACE inhibitory activity	[79]
*L. delbrueckii* subsp. *bulgaricus* IFO13953	k-cn	Ala-Arg-His-Pro-His-Pro-His-Leu-Ser-Phe-Met	Antioxidative activity	[80]
*L. rhamnosus* + digestion with pepsin and Corolase PP	β-cn	Asp-Lys-Ile-His-Pro-Phe, Tyr-Gln-Glu-Pro-Val-Leu Val-Lys-Glu-Ala-Met-Ala-Pro-Lys	ACE inhibitory activity Antioxidative activity	[81]
*L. delbrueckii* subsp. *bulgaricus*	β-cn	Ser-Lys-Val-Tyr-Pro-Phe-Pro-Gly-Pro-Ile	ACE inhibitory activity	[82]
*Streptococcus thermophilus* + *L. lactis* subsp. *lactis* biovar. diacetylactis	β-cn	Ser-Lys-Val-Tyr-Pro	ACE inhibitory activity	[82]
*L. helveticus* ICM 1004 cell-free extract	Skim milk hydrolysate	Val-Pro-Pro, Ile-Pro-Pro	ACE inhibitory activity	[83]

^a^ Abbreviations: cn = casein, ACE = angiotensin I-converting enzyme.

## Data Availability

Not applicable.

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
