# Peer review of "The Role of Bioactive Peptides in Diabetes and Obesity"

_foods, 2021, doi:10.3390/foods10092220_

Round 1

Reviewer 1 Report

Manuscript entitled "Role of bioactive peptides on diabetes and obesity" is very interesting and important, as diabetes mellitus and obesity are worldwide health problem of 21st century. But, I have few suggestions.

  1. I think that this article is too long. For readers, 45 pages may be to much. I think, that it will be more friendly for readers, if this manuscript will be divided on 2 articles.
  2. In several cases, used points must be excluded. For example, page 2 lines 53 and 54, there are G.I. It should be GI, line 59 A.C.E. > ACE, line 62 B.A.P. > BAP, and also in several other places, such as R.O.S. (page 7, line 137), M.A.P.K. , and so on. These must be changed. Also in sequences of aa, as for example R.V.P.S.L. or Y.Y. (Table 1).
  3. In description of Table 1 there are: A.C.E. , S.O.D. and so on. Write please without bold.
  4. There are Figures 1 (A), 1(B) and so on. I should be Figures 1, 2, 3 and...
  5. Figures need descriptions. There are only figures but for readers it will be difficult to understand their content.
  6. There are errors in cited references. For example, page 12 it is "Miner-Williams et al. Is it reference 52? Page 12, line 294 (Wang et al), line 297 (Courtine et al) - lack in References.
  7. Page 13 lines 314, 317, 323 and in other lines and tables, it is used Lb. What is this. It shluld be, for example L. helveticus, L. rhamnosus (not Lb.)
  8. In table 2 Lc. I don't know, what is this? I think, here should be L. lactis.
  9. In table 5, first column is unclear. For example Egg-derived peptides [75], or 264.7 RAW, macrophages, and so on. The content of this column should be described more detailed.
  10. Tables sould be changed, because they are too long (table 5 contains 9 pages), and must be used 1 font of letters (there are used different).

Author Response

Manuscript entitled "Role of bioactive peptides on diabetes and obesity" is very interesting and important, as diabetes mellitus and obesity are worldwide health problem of 21st century. But, I have few suggestions.

The authors were thankful for the reviewer comments to make the manuscript in a suitable readable script

I think that this article is too long. For readers, 45 pages may be to much. I think, that it will be more friendly for readers, if this manuscript will be divided on 2 articles.

The authors are great full for the reviewer valuable suggestion, due to the tables and figure the manuscript appears longer and the role of bioactive peptides for the treatment of diabetes and obesity. The mechanism of obesity and diabetes were correlated with each other. Hence if we separate the content information will be diluted.

In several cases, used points must be excluded. For example, page 2 lines 53 and 54, there are G.I. It should be GI, line 59 A.C.E. > ACE, line 62 B.A.P. > BAP, and also in several other places, such as R.O.S. (page 7, line 137), M.A.P.K. , and so on. These must be changed. Also in sequences of aa, as for example R.V.P.S.L. or Y.Y. (Table 1).

As per the reviewers valuable suggestion, the changes has been amended throughout the manuscript and the changes has been highlighted

In description of Table 1 there are: A.C.E., S.O.D. and so on. Write please without bold.

As per the reviewers valuable suggestion, the changes has been amended throughout the manuscript and the changes has been highlighted

There are Figures 1 (A), 1(B) and so on. I should be Figures 1, 2, 3 and...

As per the reviewers valuable suggestion, the numbering in the Figures has been edited throughout the manuscript and the changes has been highlighted

Figures need descriptions. There are only figures but for readers it will be difficult to understand their content.

As per the reviewers valuable suggestion, the detailed description has been provide for the Figures and further the changes has been highlighted

There are errors in cited references. For example, page 12 it is "Miner-Williams et al. Is it reference 52? Page 12, line 294 (Wang et al), line 297 (Courtine et al) - lack in References.

The missing reference part has been edited.

Page 13 lines 314, 317, 323 and in other lines and tables, it is used Lb. What is this. It should be, for example L. helveticus, L. rhamnosus (not Lb.)

As per the reviewers valuable suggestion, the changes has been effected in the manuscript and further highlighted

In table 2 Lc. I don't know, what is this? I think, here should be L. lactis.

The changes has been effected in the manuscript as per the reviewer valuable suggestion

In table 5, first column is unclear. For example Egg-derived peptides [75], or 264.7 RAW, macrophages, and so on. The content of this column should be described more detailed.

The changes has been effected in the manuscript as per the reviewer valuable suggestion

Tables should be changed, because they are too long (table 5 contains 9 pages), and must be used 1 font of letters (there are used different).

The changes such as font size was adjusted and further the table 5, 6 and 7 were made into supplementary files has been effected in the manuscript as per the reviewer valuable suggestion

Reviewer 2 Report

In this manuscript, Chelliah and colleagues present a comprehensive review on the biological function of bioactive peptides from three aspects, namely source, effect and diversity, and describe their potential application as metabolic drug on obesity and type 2 diabetes.

This review has provided a deep insight into bioactive peptide’s protective role. 

Minor: 

  1. The order of tables is a mess, and table 3 is missing;
  2. Table 1 and table 5 take too much space, the author should find a more compact way to present;
  3. Figure 3(C) should be Figure 3(B).

Author Response

In this manuscript, Chelliah and colleagues present a comprehensive review on the biological function of bioactive peptides from three aspects, namely source, effect and diversity, and describe their potential application as metabolic drug on obesity and type 2 diabetes.

This review has provided a deep insight into bioactive peptide’s protective role. 

The authors were grateful for the reviewer comments to make the manuscript in a suitable readable script

Minor: 

The order of tables is a mess, and table 3 is missing;

As per the reviewers valuable suggestion, the detailed description has been provide for the tables and table order has been aligned and further the changes has been highlighted

Table 1 and table 5 take too much space, the author should find a more compact way to present;

The changes such as font size was adjusted and further the table 5, 6 and 7 were made into supplementary files has been effected in the manuscript as per the reviewer valuable suggestion

Figure 3(C) should be Figure 3(B).

As per the reviewers valuable suggestion, the numbering in the Figures has been edited throughout the manuscript and the changes has been highlighted

Round 2

Reviewer 1 Report

References need major corrections:

1) References 11-27, 63-72, 75-79, 81-86, 91-104, 107-113 - lack names of journals.

2) All words in journal's name must be write using capital letter.

Author Response

The authors were thankful for the reviewer comments to make the manuscript in a suitable readable script

1) References 11-27, 63-72, 75-79, 81-86, 91-104, 107-113 - lack names of journals.

The authors are great full for the reviewer valuable suggestion, the references has been edited.

2) All words in journal's name must be write using capital letter.

As per the reviewers valuable suggestion, the changes has been amended in the manuscript and the changes has been highlighted.